# One Policy to Run Them All: Towards an End-to-end Learning Approach to Multi-Embodiment Locomotion

Nico Bohlinger*, Grzegorz Czechmanowski[†‡], Maciej Krupka[†], Piotr Kicki[†‡]
Krzysztof Walas[†‡], Jan Peters[*§] and Davide Tateo*
*Department of Computer Science, Technical University of Darmstadt, Germany
[†]Institute of Robotics and Machine Intelligence, Poznan University of Technology, Poland
[‡]IDEAS NCBR, Warsaw, Poland
[§]German Research Center for AI (DFKI), Hessian.AI, Centre for Cognitive Science

*Abstract*—The field of legged robotics is still missing a single learning framework that can control different embodiments—such as quadruped, humanoids, and hexapods—simultaneously and possibly transfer, zero or few-shot, to unseen robot embodiments. To close this gap, we introduce URMA, the Unified Robot Morphology Architecture. Our framework brings the end-to-end Multi-Task Reinforcement Learning approach to the realm of legged robots, enabling the learned policy to control any type of robot morphology. Our experiments show that URMA can learn a locomotion policy on multiple embodiments that can be easily transferred to unseen robot platforms in simulation and the real world.

## I. INTRODUCTION

The robotics community has mastered the problem of robust gait generation in the last few years. With the help of Deep Reinforcement Learning (DRL) techniques, legged robots can show impressive locomotion skills. There are numerous examples of highly agile locomotion with quadrupedal robots [20, 19, 6, 3, 32, 5], learning to run at high speeds, jumping over obstacles, walking on rough terrain, performing handstands, and completing parkour courses. Achieving these agile movements is often enabled by training in many parallelized simulation environments and using carefully tuned or automatic curricula on the task difficulty [23, 18]. Even learning simple locomotion behaviors directly on real robots is possible but requires far more efficient learning approaches [26, 27]. Similar methods have been applied to generate robust walking gaits for bipedal and humanoid robots [25, 14, 22]. The learned policies can be effectively transferred to the real world and work in all kinds of terrain with the help of extensive Domain Randomization (DR) [21, 4] during training. Additionally, techniques like student-teacher learning [20, 13] or the addition of model-based components [10, 11] or constrains [16, 12, 8] to the learning process can further improve the learning efficiency and robustness of the policies. However, the long-term objective would be to develop foundation models for locomotion, allowing zero-shot (or few-shot) deployment to any arbitrary platform. To reach this objective, it is fundamental to adapt the underlying learning system to support different tasks and morphologies.

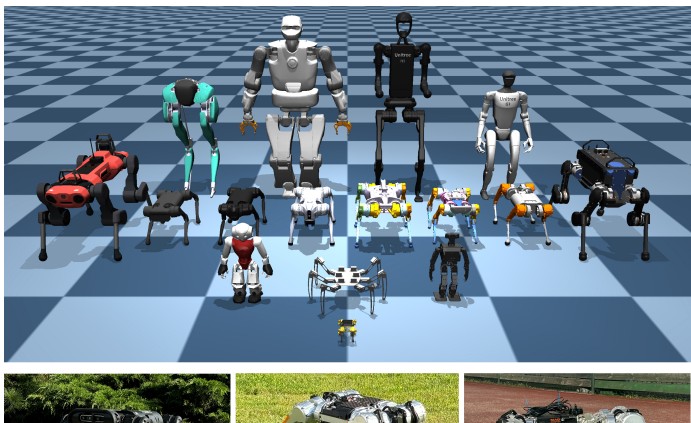

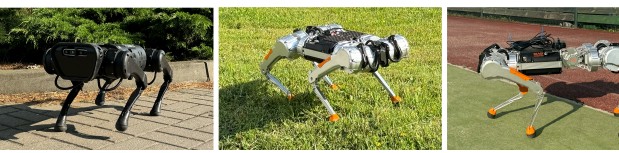

Fig. 1. Top – We train a single locomotion policy for multiple robot embodiments in simulation. Bottom – We can transfer and deploy the policy on three real-world platforms by randomizing the embodiments and environment dynamics during training.

To map differently sized observation and action spaces into and out of a shared representation space, implementations often resort to padding the observations and actions with zeros to fit a maximum length [31] or to using a separate neural network head for each task [7]. These methods allow for efficient training but can be limiting when trying to transfer to new tasks or environments: for every new robot, the training process has to be repeated from scratch, as different embodiments require different hyperparameters, reward coefficients, training curricula, etc. To tackle this problem we introduce a novel neural network architecture that can handle differently sized action and observation spaces, allowing the policy to adapt easily to diverse robot morphologies. Our method allows us zero-shot deployment of the policy to unseen robots and few-shot fine-tuning on novel target platforms.

*Related work:* Early work on controlling different robot morphologies is based on the idea of using Graph Neural Networks (GNNs) to capture the morphological structure of

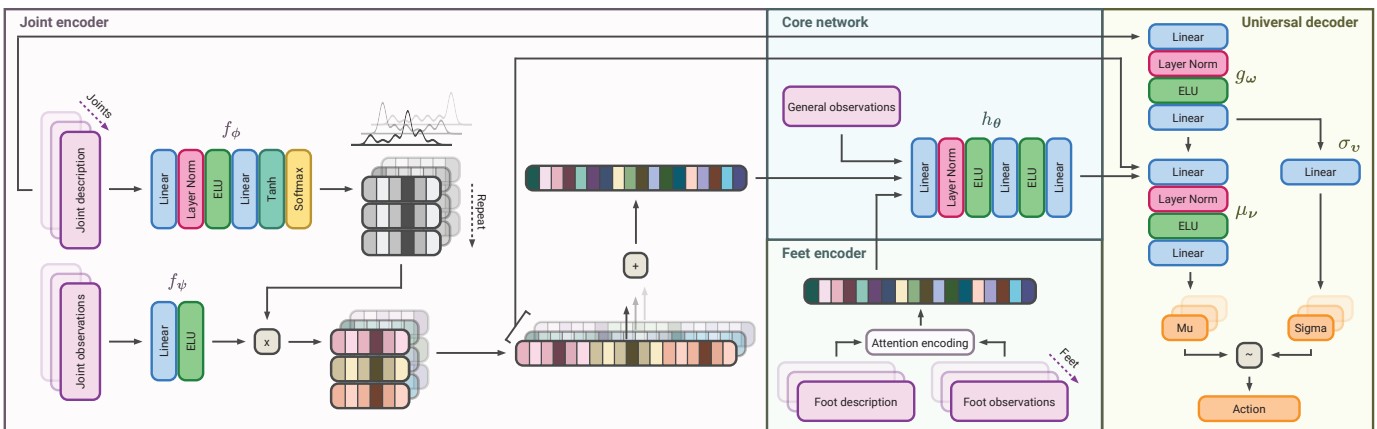

Fig. 2. Overview of the URMA architecture.

the robots [29, 9, 30]. GNN-based approaches can control different robots even when removing some of their limbs, but they struggle to generalize to many different morphologies. Transformer-based architectures have been proposed to overcome the limitations of GNNs by using the attention mechanism to globally aggregate information of varying numbers of joints [15, 28]. These methods still lack substantial generality as they are limited to morphologies that were defined a priori. Recently Shafiee et al. [24] showed that a single controller can be trained to control 16 different 3D-simulated quadrupedal robots and to transfer to two of them in the real world. Compared to all the other approaches, our method can handle multiple embodiments from any legged morphology and adapt to arbitrary joint configurations with the same network.

## II. MULTI-EMBODIMENT LOCOMOTION WITH A SINGLE POLICY

In Multi-Task Reinforcement Learning (MTRL) the objective is to learn a single policy $\pi_\theta$ that optimizes the average of the expected discounted return $\mathcal{J}^m(\boldsymbol{\theta})$ over the reward function $r^m$ across $M$ tasks:

$$\mathcal{J}(\boldsymbol{\theta}) = \frac{1}{M}\sum_m^M \mathcal{J}^m(\boldsymbol{\theta}), \quad \mathcal{J}^m(\boldsymbol{\theta}) = \mathbb{E}_{\tau \sim \pi}\left[\sum_{t=0}^T \gamma^t r^m(s,a)\right]. \quad (1)$$

In our case, we consider different robot embodiments as separate tasks and train a policy controlling all robots and optimizing the objective described in (1). To solve this problem in the multi-embodiment setting, we propose the Unified Robot Morphology Architecture (URMA), a complete morphology agnostic architecture, that does not require defining the possible morphologies or joints beforehand. Figure 2 presents a schematic overview of URMA. In general, URMA splits the observations of a robot into distinct parts, encodes them with a simple attention encoder [2] with a learnable temperature [17], and uses our universal morphology decoder to obtain the actions for every joint of the robot.

To handle observations of any morphology, URMA first splits the observation vector $o$ into robot-specific and general observations $o_g$, where the former can be of varying size, and the latter has a fixed dimensionality. For locomotion, we subdivide the robot-specific observations into joint and feet-specific observations. This split is not necessary but makes the application to locomotion cleaner. In the following text, we describe everything w.r.t. the joint-specific observations, but the same applies to the feet-specific ones as well. Every joint of a robot is composed of joint-specific observations $o_j$ and a description vector $d_j$, which is a fixed-size vector that can uniquely describe the joint. The description vectors and joint-specific observations are encoded separately by the Multilayer Perceptrons (MLPs) $f_\phi$ and $f_\psi$ and are then passed through a simple attention head, with a learnable temperature $\tau$ and a minimum temperature $\epsilon$, to get a single latent vector

$$\bar{z}_{\text{joints}} = \sum_{j\in J} z_j, \qquad z_j = \frac{\exp\left(\frac{f_\phi(d_j)}{\tau+\epsilon}\right)}{\sum_{j\in J}\exp\left(\frac{f_\phi(d_j)}{\tau+\epsilon}\right)} f_\psi(o_j), \quad (2)$$

that contains the information of the joint-specific observations of all joints. With the help of the attention mechanism, the network can learn to separate the relevant joint information and precisely route it into the specific dimensions of the latent vector by reducing the temperature $\tau$ of the softmax close to zero. The joint latent vector $\bar{z}_{\text{joints}}$ is then concatenated with the feet latent vector $\bar{z}_{\text{feet}}$ and the general observations $o_g$ and passed to the policies core MLP $h_\theta$ to get the action latent vector $\bar{z}_{\text{action}} = h_\theta(o_g, \bar{z}_{\text{joints}}, \bar{z}_{\text{feet}})$. To obtain the final action for the robot, we use our universal morphology decoder, which takes the general action latent vector and pairs it with the set of encoded specific joint descriptions and the single joint latent vectors to produce the mean and standard deviation of the actions for every joint, from which the final action is sampled as

$$a^j \sim \mathcal{N}(\mu_{\boldsymbol{\nu}}(d_j^a, \bar{z}_{\text{action}}, z_j), \sigma_{\boldsymbol{\nu}}(d_j^a)), \qquad d_j^a = g_{\boldsymbol{\omega}}(d_j). \quad (3)$$

To ensure that only fully normalized and well-behaved observations come into the network, we use LayerNorm [1] after every input layer. The learning process also benefits from adding another LayerNorm in the action mean network $\mu_{\boldsymbol{\nu}}$. We argue that this choice improves the alignment of the different latent vectors entering into $\mu_{\boldsymbol{\nu}}$ better. To ensure a fair

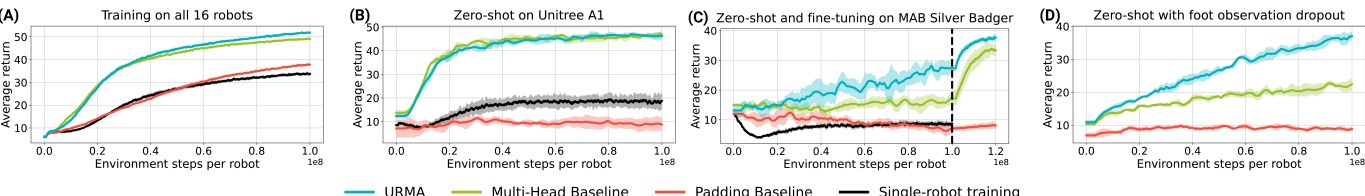

Fig. 3. (A) – Average return of the three architectures during training on all 16 robots compared to the single-robot training setting. (B) – Zero-shot transfer to the Unitree A1 while training on the other 15 robots. (C) – Zero-shot transfer to the MAB Robotics Silver Badger while training on the other 15 robots and fine-tuning on only the Silver Badger afterward. (D) – Zero-shot evaluation on all 16 robots while removing the feet observations.

comparison, we also use LayerNorms with the same rationale in the baseline architectures.

*Results*

First, we want to evaluate the training efficiency of MTRL in our setting. We train URMA and the baselines on all 16 robots in the training set simultaneously and compare the average return to the single-robot training setting, where a separate policy is trained for every robot. All policies are trained on 100 million steps per robot, and every experiment shows the average return over 5 seeds and the corresponding 95% confidence interval. Figure 3 confirms the advantage in learning efficiency of MTRL over single-task learning, as URMA and the multi-head baseline learn significantly faster than training only on a single robot at a time. It has to be noted that in the multi-task training, we set a fixed batch size per robot, so 16-times this batch size corresponds to the total PPO batch size. In the single-robot training, we only use the fixed batch size for the single robot. This lower effective batch size in the single-robot training leads to its worse performance compared to the multi-task setting. Furthermore, early on in training, URMA learns slightly slower than the multi-head baseline due to the time needed by the attention layers to learn to separate the robot-specific information, which the multi-head baseline inherently does from the beginning. However, URMA ultimately reaches a higher final performance. The padding baseline performs noticeably worse than the other two. We argue that the policy has trouble learning the strong separation in representation space between the different robots—which is necessary for the differently structured observation and action spaces—only based on the task ID.

Next, we evaluate the zero-shot and few-shot transfer capabilities of URMA and the baselines on two robots that were withheld from the training set of the respective policies. We test the zero-shot transfer on the Unitree A1, a robot whose embodiment is similar to other quadrupeds in the training set. Figure 3 shows the evaluation for the A1 during a training process with the other 15 robots and highlights that both URMA and the multi-head baseline can transfer perfectly well to the A1 while never having seen it during training. The policy only trained specifically on the A1 (shown in black) performs distinctly worse since the DR for all quadruped robots is particularly hard to ensure transferability to the real platforms. Likely, the total batch size in the single-robot setting is again too small to learn the task well. Furthermore,

the time-dependent reward curriculum can also cause dips in performance during the early learning phase if the policy cannot keep up with the increasing penalty coefficients.

To investigate an out-of-distribution embodiment, we use the same setup as for the A1 and evaluate zero-shot on the MAB Robotics Silver Badger robot, which has an additional spine joint in the trunk and lacks feet observations, and then fine-tune the policies for 20 million steps only on the Silver Badger itself. The results show that URMA can handle the additional joint and the missing feet observations better than the baselines and is the only method capable of achieving a good gait at the end of training. After starting the fine-tuning, URMA maintains the lead in the average return due to the better initial zero-shot performance. To further assess the adaptability of our approach, we evaluate the zero-shot performance in the setting where observations are dropped out, which can easily happen in real-world scenarios due to sensor failures. To test the additional robustness in this setting, we train the architectures on all robots with all observations and evaluate them on all robots while completely dropping the feet observations. Figure 3 confirms the results from the previous experiment and shows that URMA can handle missing observations better than the baselines.

Finally, we deploy the same URMA policy on the real Unitree A1, MAB Honey Badger, and MAB Silver Badger quadruped robots. Figure 1 shows the robots walking with the learned policy on pavement, grass, and plastic turf terrain with slight inclinations. Due to the extensive DR during training, the single policy can be zero-shot transferred to the three real robots without any further fine-tuning. While the Unitree A1 and the MAB Silver Badger are in the training set, the network is not trained on the MAB Honey Badger. Despite the Honey Badger's gait not being as good as the other two robots, it can still locomote robustly on the terrain we tested, proving the generalization capabilities of our architecture and training scheme.

*Limitations:* While our method is the first end-to-end approach for learning multi-embodiment locomotion, many open challenges remain. On one side, our generalization capabilities rely mostly on the availability of data, therefore zero-shot transfer to embodiments that are completely out of the training distribution is still problematic. This issue could be tackled by exploiting other techniques in the literature, such as data augmentation and unsupervised representation learning, to improve our method's generalization capabilities. Furthermore,

we currently omit exteroceptive sensors from the observations, which can be crucial to learning policies that can navigate in complex environments and fully exploit the agile locomotion capabilities of legged robots.

## III. CONCLUSION

We presented URMA, a framework to learn robust locomotion for different types of robot morphologies end-to-end with a single neural network architecture. Our flexible learning framework and the efficient encoders and decoders allow URMA to learn a single control policy for 16 different embodiments from three different legged robot morphologies. In practice, URMA reaches higher final performance on the training with all robots, shows higher robustness to observation dropout, and better zero-shot capabilities to new robots compared to MTRL baselines. Furthermore, we deploy the same policy zero-shot on two known and one unseen quadruped robot in the real world. We argue that this multi-embodiment learning setting can be easily extended to more complex scenarios and can serve as a basis for locomotion foundation models that can act on the lowest level of robot control.

## ACKNOWLEDGMENTS

This project was funded by National Science Centre, Poland under the OPUS call in the Weave programme UMO-2021/43/I/ST6/02711, and by the German Science Foundation (DFG) under grant number PE 2315/17-1. Part of the calculations were conducted on the Lichtenberg high performance computer at TU Darmstadt.

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
