# OpenReview forum: "One Policy to Run Them All: Towards an End-to-end Learning Approach to Multi-Embodiment Locomotion"
_roboticsfoundation.org/RSS/2024/Workshop/EARL — EARL 2024 Poster_

### Official Review · Reviewer_eYSZ · 2024-06-24
**An End-to-end Learning Approach to Multi-Embodiment Locomotion**

**Rating:** 7
**Confidence:** 3

**Review:**

Summary: The paper presents URMA, which stands for Unified Robot Morphology Architecture. URMA can learn a locomotion policy on multiple embodiments that can be transferred to other similar robot platforms. The core ideas include 1) splitting the observations into robot-specific observations and general observations, 2) mapping different observations into a general latent vector using MLPs, and 3) training a controller (called universal morphology decoder) that maps the latent vector to action.

Comment:

1) It is not clear to me why learning on all robots simultaneously could outperform single-robot training. How is the average return being computed when training on with all robots and with a single robot?

2) The generalization of the policy to another robot platform, e.g., A1, is interesting. However, this is mainly because the training example contains robots that have a similar morphology.

3) I appreciate the nice visualization of the URMA architecture. However, it is difficult to understand the structure. The main issue is that it contains too many details. The input and output somehow mingle with the network details.

4) Fig 3. The caption doesn’t seem to match the figure. e.g., there is no top left or top right. I guess the authors forgot to change the caption after changing the figure layout.


Overall, this is an interesting paper.

---

### Decision · Program_Chairs · 2024-06-24

Accept (Poster)